# SH3RF2 contributes to cisplatin resistance in ovarian cancer cells by promoting RBPMS degradation

Ting-Ting Gong[1,6], Fang-Hua Liu[2,6], Qian Xiao[1,2,6], Yi-Zi Li[2], Yi-Fan Wei[2], He-Li Xu[2], Fan Cao[2], Ming-Li Sun[1], Feng-Li Jiang[1], Tao Tao[1], Qi-Peng Ma[1], Xue Qin [1✉], Yang Song[1], Song Gao[1], Lang Wu[3], Yu-Hong Zhao[2,4], Dong-Hui Huang [2,4✉] & Qi-Jun Wu [1,2,4,5✉]

Platinum-based chemotherapy remains one of the major choices for treatment of ovarian cancer (OC). However, primary or acquired drug resistance severely impairs their efficiency, thereby causing chemotherapy failure and poor prognosis. SH3 domain containing ring finger 2 (SH3RF2) has been linked to the development of cancer. Here we find higher levels of SH3RF2 in the tumor tissues from cisplatin-resistant OC patients when compared to those from cisplatin-sensitive patients. Similarly, cisplatin-resistant OC cells also express higher levels of SH3RF2 than normal OC cells. Through in vitro and in vivo loss-of-function experiments, SH3RF2 is identified as a driver of cisplatin resistance, as evidenced by increases in cisplatin-induced cell apoptosis and DNA damage and decreases in cell proliferation induced by SH3RF2 depletion. Mechanistically, SH3RF2 can directly bind to the RNA-binding protein mRNA processing factor (RBPMS). RBPMS has been reported as an inhibitor of cisplatin resistance in OC. As a E3 ligase, SH3RF2 promotes the K48-linked ubiquitination of RBPMS to increase its proteasomal degradation and activator protein 1 (AP-1) transactivation. Impairments in RBPMS function reverse the inhibitory effect of SH3RF2 depletion on cisplatin resistance. Collectively, the SH3RF2-RBPMS-AP-1 axis is an important regulator in cisplatin resistance and inhibition of SH3RF2 may be a potential target in preventing cisplatin resistance.

[1] Department of Obstetrics and Gynecology, Shengjing Hospital of China Medical University, Shenyang, China. [2] Department of Clinical Epidemiology, Shengjing Hospital of China Medical University, Shenyang, China. [3] Cancer Epidemiology Division, Population Sciences in the Pacific Program, University of Hawaii Cancer Center, University of Hawaii at Manoa, Honolulu, HI, USA. [4] Liaoning Key Laboratory of Precision Medical Research on Major Chronic Disease, Shengjing Hospital of China Medical University, Shenyang, China. [5] NHC Key Laboratory of Advanced Reproductive Medicine and Fertility (China Medical University), National Health Commission, Shenyang, China. [6] These authors contributed equally: Ting-Ting Gong, Fang-Hua Liu, Qian Xiao. ✉email: qinx@sj-hospital.org; huangdh_cc@163.com; wuqj@sj-hospital.org

Ovarian cancer (OC) is the eighth most common malignancies and the eighth leading cause of cancer-related mortality among females globally with 313,959 new cases and 207,252 new deaths in 2020[1]. Early diagnosis of OC is difficult due to the lack of obvious symptoms. Accordingly, 70% of patients are diagnosed at an advanced stage for OC, and the five-year survival rate is still less than 50% due to late diagnosis[2]. In China, OC has replaced uterine cancer and became the 2nd leading cause of mortality in gynecological cancer[3]. It is estimated that OC burden in China would continue to rise with a higher rate in the next decade[4].

Chemotherapy remains one of the primary therapies for cancer treatment. Platinum is one of the first-line antineoplastic drugs for OC treatment. Cisplatin (DDP) is able to interact with DNA and form platinum-DNA adducts, and thereby inhibits cell proliferation and activates DNA damage response and the pro-apoptotic signaling pathway[5]. Unfortunately, chemoresistance is continually encountered in clinical practice. Chemoresistance can be acquired in numerous ways; it can be developed from increased activities of anti-apoptotic signaling pathways, activation of repair, detoxification mechanisms, or genetic and epigenetic variations in cancer cells[6,7]. The development of chemoresistance in cancer patients limits the clinical effectiveness of chemotherapy and triggers local infiltration and distant metastases. Therefore, it is critical to understand the molecular mechanism of OC chemoresistance.

SH3 domain containing ring finger 2 (SH3RF2, also referred to as POSHER) is located on Chromosome 5q32 in the human genome and it encodes a ubiquitin E3 ligase which contains three SH3 domains and a ring finger domain[8]. Emerging studies demonstrated that SH3RF2 was involved in the development of nonalcoholic fatty liver disease[9] and autism spectrum disorders[10]. Kim et al. [11] suggested that SH3RF2 effectively inhibited cell apoptosis and promoted cell migration, cell proliferation, and xenograft tumor growth. So far, the function of SH3RF2 has been poorly characterized in chemoresistance.

In several recent studies, RNA binding proteins were found to be implicated in tumor progression and therapy resistance in OC[12,13]. RNA binding protein, mRNA processing factor (RBPMS, also referred to as HERMES) is located on chromosome 8p12 in the human genome[14]. RBPMS has a single RNA recognition motif that consists of 23-amino acid N-terminal and 95-amino acid C-terminal regions[15]. RBPMS plays a multifunctional role in biological process; it is capable of modulating RNA splicing, transport, and stability[16,17]. RBPMS serves as a tumor suppressor in OC[15,18]. Reduced RBPMS confers drug resistance to cancer cells, including bortezomib and DDP resistance[15,18]. Targeting RBPMS may be a potential strategy for modulation of DDP resistance in cancers.

Herein, we found higher expression levels of SH3RF2 in the tumor tissues from DDP-resistant OC patients and DDP-resistant OC cells. The effect of shRNA-targeted suppression of SH3RF2 on DDP resistance and DDP-induced DNA damage in DDP-resistant OC cells and in xenograft OC tumors was explored. Twenty-five proteins have been confirmed to interact with SH3RF2. RBPMS is one of the proteins that can interact with SH3RF2. The present study investigated the relationship between SH3RF2 and RBPMS and whether RBPMS was required for cellular sensitivity to DDP in OC induced by SH3RF2 inhibition.

## Results

### The level of SH3RF2 was higher in cisplatin (DDP)-resistant OC patients and in platinum-resistant OC cells.
DDP-sensitive and DDP-resistant tumor tissues of OC patients were collected and used to perform immunohistochemistry (IHC) staining using anti-SH3RF2. As shown in Fig. 1a and b, higher expression of SH3RF2 was observed in the DDP-resistant group. Subsequently, the correlation between SH3RF2 expression levels and clinicopathological parameters of OC patients was analyzed (Table 1). Results showed that high expression of SH3RF2 was significantly associated with tumor sizes and DDP resistance.

DDP-resistant OC cells (A2780 and SKOV3) were established by exposure to different concentration of DDP (2, 4, 8, 16, 32, and 64 μM) as shown in Fig. 2a. The results of 3-(4,5-dimethylthiazol-2-yl)−2,5 diphenyl tetrazolium bromide (MTT) assays showed that the IC50 value of DDP in DDP-resistant A2780 and SKOV3 cells was higher, when compared with their parental cells (Fig. 2b and c). SH3RF2 mRNA and protein expression levels were higher in the DDP-resistant OC cells than those in the parental OC cells (Fig. 2d and e). Subsequently, carboplatin (CP)-resistant A2780 and SKOV3 cells were used to evaluate the expression of SH3RF2. CP-resistant cells also showed a higher $IC_{50}$ value of CP (Supplementary Fig. 1a and b) and increased SH3RF2 mRNA and protein expression levels (Supplementary Fig. 1c and d). The findings indicated that up-regulated SH3RF2 expression might be associated with platinum resistance in OC.

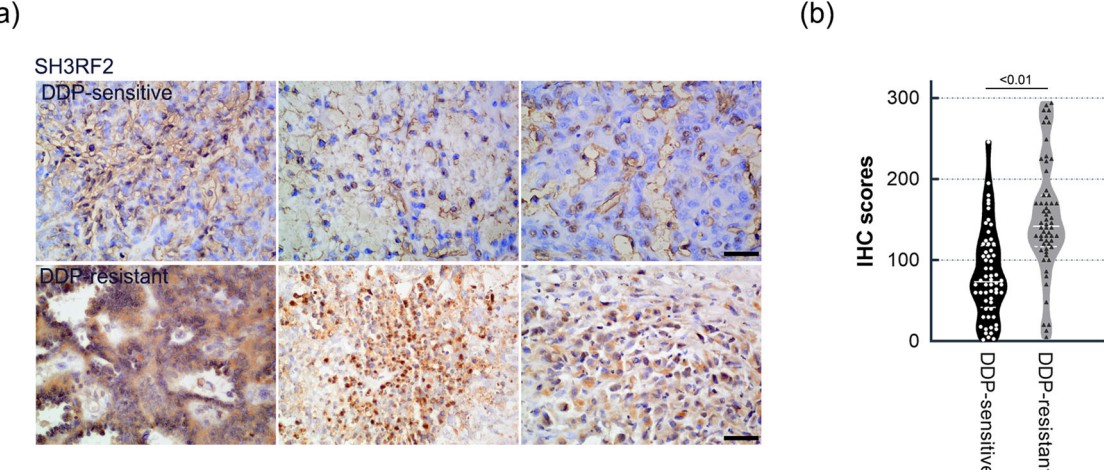

**Fig. 1 Expression of SH3RF2 in the tumor tissues from DDP-sensitive and DDP-resistant OC patients. a** Representative images of IHC staining of SH3RF2 in the tumor tissues from DDP-sensitive and DDP-resistant OC patients (Scale bars, 50 μm). **b** The quantification of SH3RF2 staining was performed by two pathologists. $n = 60$. The $p$ values were determined by Mann-Whitney U test.

**Table 1 Correlation between SH3RF2 expression levels and clinicopathological parameters of OC patients.**

| Clinical parameters | Cases | SH3RF2 expression levels | | p value |
|---|---|---|---|---|
| | | High | Low | |
| Overall | 120 | 61 | 59 | |
| Age | | | | 0.2031[a] |
| ≤55 | 62 | 35 | 27 | |
| >55 | 58 | 26 | 32 | |
| FIGO stage | | | | 0.2716[a] |
| I-II | 18 | 7 | 11 | |
| III-IV | 102 | 54 | 48 | |
| Tumor size | | | | <0.05[a] |
| ≤7 cm | 66 | 28 | 38 | |
| >7 cm | 54 | 33 | 21 | |
| Metastasis | | | | 0.3029[a] |
| Positive | 98 | 52 | 46 | |
| Negative | 22 | 9 | 13 | |
| Grading | | | | 0.4561[b] |
| High-grade | 111 | 58 | 53 | |
| Low-grade | 9 | 3 | 6 | |
| Subtype | | | | 0.8269[a] |
| Serous | 98 | 51 | 47 | |
| Mucinous | 6 | 3 | 3 | |
| Others | 16 | 7 | 9 | |
| DDP-based chemotherapy | | | | <0.0001[a] |
| DDP resistant | 61 | 46 | 15 | |
| DDP sensitive | 59 | 15 | 44 | |

[a]Chi-square test.
[b]Chi-square test with Yates' correction.

**SH3RF2 knockdown enhanced DDP efficiency in OC in vitro and in vivo**. To assess the effect of SH3RF2 on the chemosensitivity of OC cells to DDP, we silenced SH3RF2 in DDP-resistant A2780 and SKOV3 cell lines. As shown in Fig. 3a, two shRNAs (sh1-SH3RF2 and sh2-SH3RF2) decreased SH3RF2 mRNA and protein expression in both cell lines compared to the shNC. The $IC_{50}$ values of DDP in the DDP-resistant cells were notably reduced by SH3RF2 silencing (Fig. 3b). SH3RF2 silencing induced cell apoptosis and further increased DDP-induced apoptosis in DDP-resistant OC cells, as determined by flow cytometry (Fig. 3c) and acridine orange (AO)/ethidium bromide (EB) staining (Fig. 3d). Cell apoptosis was also validated by detecting cleaved PARP and cleaved caspase 3 expressions and caspase 3 activities. The level of cleaved PARP and cleaved caspase 3 expressions (Fig. 3e) and the activities of caspase 3 (Fig. 3f) were elevated in SH3RF2-silenced cells with or without DDP treatment. SH3RF2 silencing significantly reduced colony formation and promoted the inhibitory effect of DDP on colony formation compared to the negative control (Fig. 3g). Cisplatin resistance is often related to cancer stemness. Stem-like properties were evaluated by sphere formation assays. The stem-like properties of DDP-resistant A2780 cells were significantly reduced by SH3RF2 silencing (Supplementary Fig. 2). SH3RF2 silencing had the tendency to decrease the stem-like properties of DDP-resistant SKOV3 cells (Supplementary Fig. 2). These findings suggested that SH3RF2 silencing promoted the chemosensitivity of OC cells to DDP.

We analyzed the effect of SH3RF2 on DNA damage induced by DDP in OC cells by performing comet assays and detecting γH2AX levels using immunofluorescence staining and western blotting. According to the comet assay (Fig. 4a), there was an increase in DDP-induced DNA damage in SH3RF2-silenced cells compared to the negative control. The expression of the marker for DNA double-strand breaks, γH2AX, was up-regulated by SH3RF2 silencing (Fig. 4b and c). These findings indicated that SH3RF2 silencing enhanced DNA damage induced by DDP.

Next, we detected the effect of SH3RF2 on DDP resistance in OC in vivo. The inhibition in the volume and weight of xenograft OC tumors induced by DDP was further facilitated by SH3RF2 silencing (Fig. 5a and b). The expression of SH3RF2 and the markers of cell proliferation (Ki67), cell apoptosis (cleaved caspase 3), and DNA damage (γH2AX) were detected by IHC staining (Fig. 5c–e). Compared to those in the DDP+shNC group, the tumor tissues in the DDP+sh-SH3RF2 group showed a weaker staining for SH3RF2 and Ki67 but a stronger staining for cleaved caspase 3 and γH2AX. These findings indicated that SH3RF2 silencing suppressed DDP resistance in OC.

**SH3RF2 interacted with RBPMS and hindered RBPMS-mediated repressions in the transactivation of activator protein 1 (AP-1) in OC**. As a E3 ubiquitin ligase, SH3RF2 can interact with other proteins and it is involved in protein post-translational modifications. The RING domain (Fig. 6a) is an important domain implicated in the modulation of protein degradation[19]. As shown in Fig. 6b, RBPMS was one of the proteins that have been confirmed to bind to SH3RF2. We detected the RBPMS expression in the tumor tissues from DDP-sensitive and DDP-resistant OC patients. Lower expression of RBPMS was observed in the DDP-resistant group compared to the DDP-sensitive group (Supplementary Fig. 3). There was a negative correlation between IHC scores of SH3RF2 and RBPMS staining in tumor tissues from OC patients (Fig. 6c). Colocalization of endogenous SH3RF2 and RBPMS was observed in DDP-resistant A2780 and SKOV3 cell lines (Fig. 6d and Supplementary Fig. 4a). Co-immunoprecipitation assays confirmed the interaction between SH3RF2 and RBPMS in DDP-resistant A2780 and SKOV3 cell lines (Fig. 6e). Compared with the parental SKOV3 cells, the DDP-resistant SKOV3 cells showed a stronger interaction between SH3RF2 and RBPMS (Supplementary Fig. 4b). Notably, an increase in the levels of RBPMS protein expression in DDP-resistant OC cells and xenograft tumors was elicited by SH3RF2 silencing (Fig. 6f and g), implying that SH3RF2 might participate in the post-translational modification of RBPMS. Considering the function of SH3RF2, the cycloheximide (CHX) assays and ubiquitination assays were performed to analyze the effect of SH3RF2 on the stability and ubiquitination of RBPMS. SH3RF2 silencing delayed the degradation of RBPMS in DDP-resistant OC cells without MG132 (a well-known proteasome inhibitor) treatment; however, SH3RF2 silencing have no obvious effect on the degradation of RBPMS in DDP-resistant OC cells upon MG132 treatment compared to the negative control (shNC) cells upon MG132 treatment (Fig. 6h and Supplementary Fig. 4c). The findings indicated that SH3RF2 inhibited the stability of RBPMS by regulating proteasomal degradation. Next, we found that the K48-linked ubiquitination of RBPMS was suppressed by SH3RF2 silencing (Fig. 6i). The wild-type SH3RF2 or the inactive mutant (RING mutant) of SH3RF2 was overexpressed in DDP-resistant OC cells. As shown in Fig. 6j, compared with the wild-type SH3RF2, the inactive mutant of SH3RF2 had no obvious effect on the K48-linked ubiquitination of RBPMS, indicating that the catalytic effect of SH3RF2 was critical for the ubiquitination of RBPMS. These findings demonstrated that SH3RF2 mediated the K48-linked ubiquitination of RBPMS to reduce the stability of RBPMS in OC.

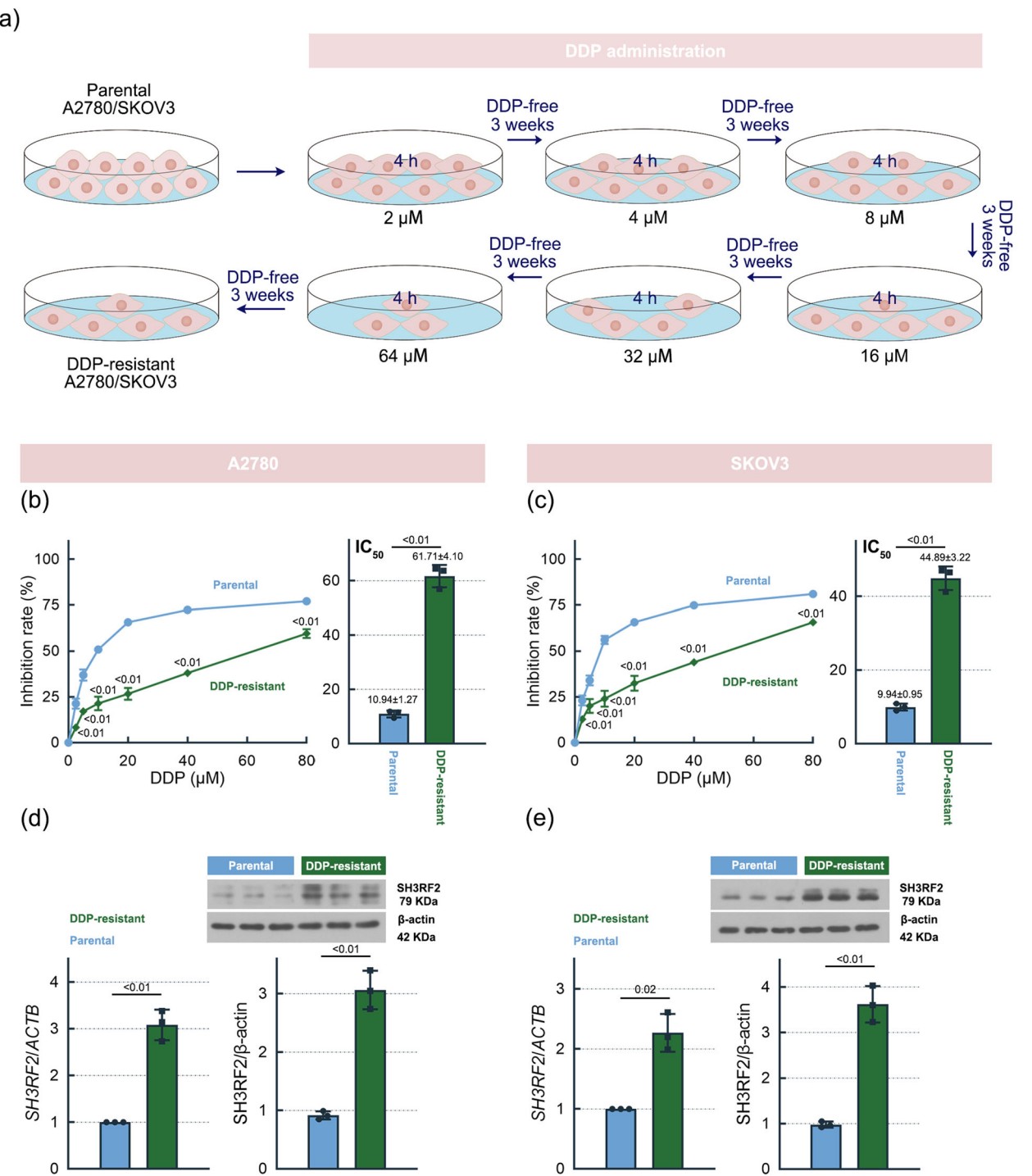

**Fig. 2 Expression of SH3RF2 in DDP-resistant OC cells. a** Schema of the establishment of DDP-resistant OC cell lines. All elements of this image were created by T-TG. **b**, **c** Parental and DDP-resistant OC cells were exposed to 0, 2.5, 5, 10, 20, 40, or 80 μM DDP for 48 h and then MTT assays were performed ($p < 0.01$ vs. the parental group). $n = 3$. The IC50 value of DDP in parental and DDP-resistant OC cells were calculated and the DDP-resistant OC cells showed significantly increased IC50 values of DDP. $n = 3$. **d**, **e** The mRNA and protein expression of SH3RF2 in parental and CP-resistant OC cells were measured by RT-qPCR ($n = 3$) and western blotting ($n = 3$). Data are expressed as the mean ± SD. *$p < 0.05$. The $p$ values were determined by two-way ANOVA, unpaired Student's t-test, or Welch's t test.

RBPMS binds to the components of AP-1, including Fos, FosB, and FosL1, and suppresses the transcriptional activity of AP-1[20]. We investigated the effect of SH3RF2 on the transcriptional activity of AP-1. SH3RF2 silencing significantly repressed the transcriptional activity of AP-1, as determined by dual luciferase reporter assays (Fig. 7a). The mRNA levels of AP-1 target genes (*XIAP* and *MYC*) were measured by RT-qPCR. A decline in *XIAP*

and *MYC* mRNA levels in DDP-resistant OC cells and xenograft tumors was triggered by SH3RF2 silencing (Fig. 7b–d), which indicated that SH3RF2 silencing resulted in repressions in AP-1 transactivation.

**The increased chemosensitivity of OC cells to DDP induced by SH3RF2 knockdown was RBPMS-dependent.** To explore the

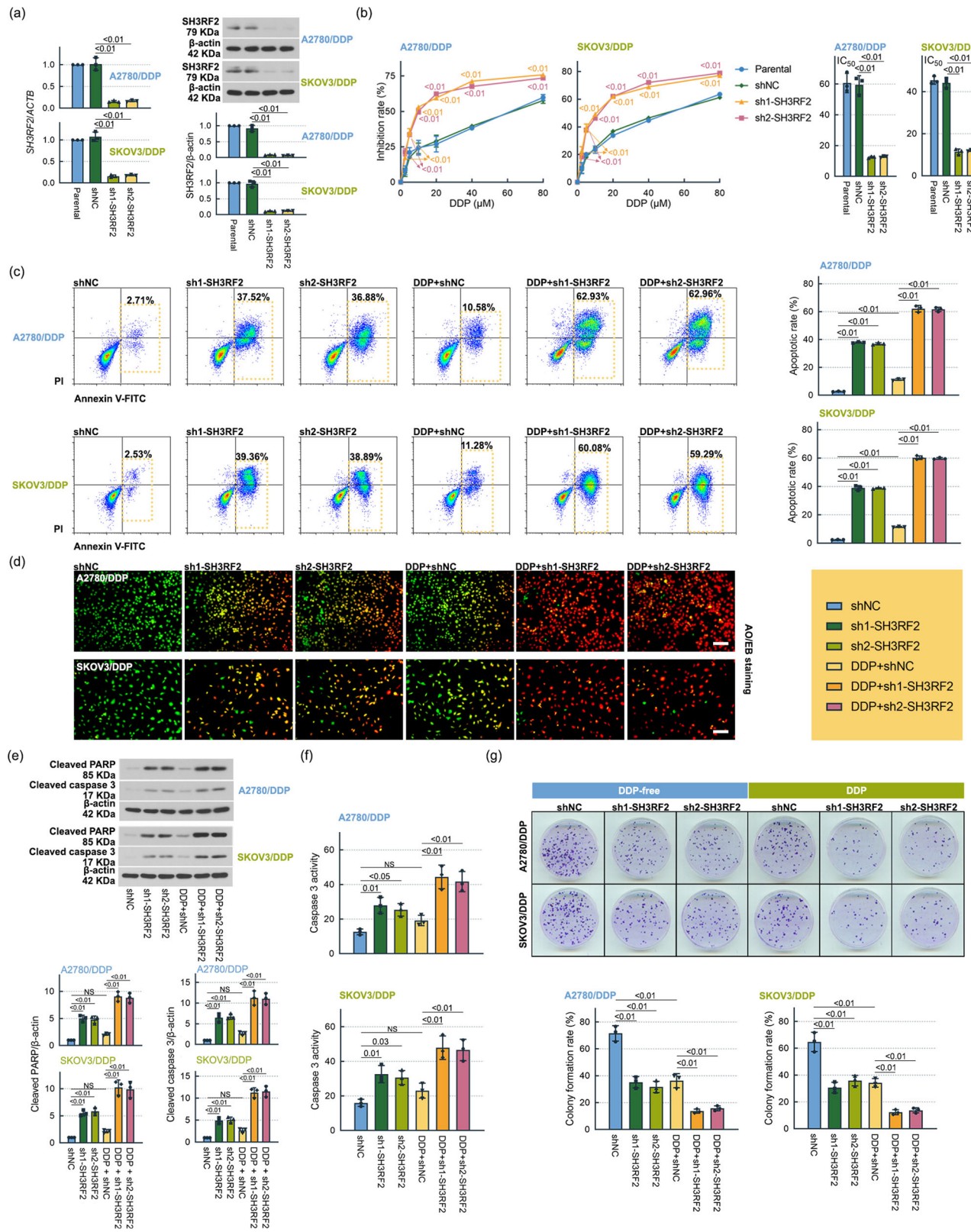

function of RBPMS in SH3RF2-mediated DDP resistance in OC cells, we silenced RBPMS in SH3RF2-silenced DDP-resistant A2780 cell line using a validated small interfering RNA (siRNA) targeting RBPMS (Fig. 8a). As shown in Fig. 8b, the siRNA decreased RBPMS protein expression in the SH3RF2-silenced DDP-resistant A2780 cell line. MTT assays showed that the decreased viability of DDP-treated

cells induced by SH3RF2 silencing was reversed by RBPMS silencing (Fig. 8c). RBPMS silencing also inhibited the promotion of cell apoptosis (Fig. 8d and e) and DNA damage (Fig. 8f and g) resulted from SH3RF2 silencing. These findings indicated that SH3RF2 was involved in DDP resistance through inhibition of RBPMS. RBPMS silencing had the tendency to reverse the inhibition of AP-1 induced

**Fig. 3 Effect of SH3RF2 depletion on DDP resistance in OC in vitro. a** DDP-resistant OC cells with stable depletion of SH3RF2 were established and the expression of SH3RF2 was evaluated by RT-qPCR ($n = 3$) and western blotting ($n = 3$). **b** DDP-resistant OC cells with or without depletion of SH3RF2 were exposed to 0, 2.5, 5, 10, 20, 40, or 80 μM DDP for 48 h and then MTT assays were performed ($p < 0.01$ vs. the shNC group). $n = 3$. The IC50 values of DDP in parental and DDP-resistant OC cells were calculated and SH3RF2 depletion significantly decreased IC50 values of DDP in DDP-resistant OC cells. $n = 3$. DDP-resistant OC cells with or without depletion of SH3RF2 were exposed to DDP (A2780/DDP: 30 μM; SKOV3/DDP: 22 μM) for 48 h. **c** Cell apoptosis were determined by flow cytometry following Annexin V-FITC/PI double staining. $n = 3$. **d** Cell apoptosis were detected by AO/EB staining (Scale bars, 100 μm). $n = 3$. **e** Western blots of lysates from DDP-resistant OC cells stained for cleaved PARP and cleaved caspase 3. $n = 3$. **f** Caspase 3 activities were measured by the commercial assay kits. $n = 3$. **g** Cell proliferation was determined by colony formation assay. $n = 3$. Data are expressed as the mean ± SD. The $p$ values were determined by one-way or two-way ANOVA. NS, no significance.

by SH3RF2 silencing (Fig. 8h), implied that SH3RF2 caused AP-1 transactivation at least partly via the inhibition of RBPMS. A schematic diagram of potential molecular mechanism underlying DDP resistance in OC was shown in Fig. 8i.

## Discussion

Chemotherapy often shows an effective initial response. After a period of chemotherapy treatment, relapse of cancer patients with resistance to drug therapy may occur, resulting in the failure of treatments in cancer patients. Platinum, alone or in combination with other chemotherapeutic drugs or other therapies, is considered as a classic management of OC[21]. Numerous molecular mechanisms of resistance to chemotherapy have been reported in OC; however, drug resistance continues to occur in OC patients. Therefore, it is imperative to continuously understand the molecular mechanism of chemotherapy resistance in OC. In the present study, we elucidated that SH3RF2 contributed to DDP resistance in OC. The findings uncovered a mechanistic basis by which SH3RF2 assisted tumor cells to reduce DNA damage, evade cell apoptosis, and promote cell proliferation induced by DDP in OC cell lines and xenograft tumor models. RBPMS, a regulator of DDP sensitivity in OC, was found to interact with SH3RF2. SH3RF2 reduced RBPMS protein stability by modulating protein degradation mediated by the ubiquitin-proteasome pathway. Mechanistically, SH3RF2 reduced DNA damage and cell apoptosis by impairing the function of RBPMS to inhibit DDP-induced DNA damage and cell apoptosis. Based on our findings, the SH3RF2-RBPMS axis may function as a potential target for boosting the efficacy of DDP-based chemotherapy.

SH3RF2 belongs to the SH3RF family of proteins that are characterized with the presence of a ring finger domain and multiple SH3 domains. In addition to SH3RF2, there are two other members, SH3RF3 and POSH, in the SH3RF family. Among the three members, SH3RF3 has been reported to modulate the proliferation, invasion, and self-renewal abilities of OC cells[22]. Up-regulated levels of SH3RF2 are observed in the tumor tissues from OC patients compared to those in the normal tissues (Supplementary Fig. 5); Here we identified SH3RF2 up-regulation in DDP-resistant OC patients and platinum-resistant OC cell lines and reported a previously unidentified role of SH3RF2 in DDP resistance silence in OC. A previous study has uncovered the capability of SH3RF2 in suppressing cell apoptosis[23]. We found that shRNA-mediated depletion of SH3RF2 could exert anti-tumor effects in DDP-resistant OC cells and xenograft tumor models through promoting cell apoptosis and inhibiting cell proliferation. As expected, the depletion of SH3RF2 restored the sensitivity of DDP-resistant OC cell lines to DDP, indicating that SH3RF2 upregulation served as a potent contributor to DDP resistance. These results suggest that targeting SH3RF2 may be a potential strategy to improve the response to platinum-based chemotherapy.

Protein degradation plays a vital role in maintaining cellular homeostasis, and abnormal accumulation of proteins may be harmful to cells and lead to numerous diseases, including cancers[24]. Proteasome-mediated degradation and lysosomal-mediated proteolysis are considered as major proteolytic pathways in eukaryotes. Approximately 80%-90% of intracellular protein destruction occurs via the ubiquitin-proteasome system, thus the ubiquitin-proteasome system is involved in regulation of biological processes, including cell proliferation and apoptosis[25]. The degradation mediated by ubiquitin-proteasome system consists of two discrete steps: (i) a single ubiquitin or multiple ubiquitin molecules is transferred to the protein substrate via E1 (ubiquitin-activating enzymes), E2 (ubiquitin-conjugating enzymes), and E3 (ubiquitin ligases) enzymes; (ii) the ubiquitinated protein is degraded by the 26 S proteasome complex[26]. As an E3 ligase, SH3RF2 can induce the proteasomal degradation of its target proteins[9,23]. We found that a previously reported regulator of DDP resistance in OC, RBPMS, is an interacting protein of SH3RF2. The previous study reported that the levels of RBPMS significantly decreased in OC cells after acquiring DDP resistance and decreased RBPMS caused reductions in cellular sensitivity to DDP[15]. In this study, we investigated whether RBPMS functioned as a downstream target of SH3RF2 in OC. We confirmed that SH3RF2 could bind to RBPMS in OC cells. Our findings uncovered that SH3RF2 reduced RBPMS protein stability in OC cells by regulating its proteasomal degradation. Ubiquitin can be conjugated by another ubiquitin through its internal Lys residues (K6, K11, K27, K29, K33, K48, and K63) or through its N-terminal methionine[27]. K48-linkage ubiquitination is linked to proteasomal degradation. SH3RF2 mediates the degradation of its substrate in a ubiquitin-proteasome-dependent manner via K48-linkage ubiquitination[9]. In our study, SH3RF2 was found to mediate RBPMS degradation in OC cells through a K48-linked ubiquitination. We also confirmed that SH3RF2 conferred DDP resistance in OC cells through promoting the proteasomal degradation of RBPMS.

AP-1, a family of transcription factors, is composed of heterodimers of a JUN family member and a FOS family member[28]. AP-1 has been linked to therapy resistance, including chemotherapy resistance[29,30], endocrine therapy resistance[31], and radiotherapy resistance[32]. RBPMS directly interacts with c-FOS to inhibit the binding between c-FOS and c-JUN, thus decreasing the transcriptional activity of AP-1; RBPMS also interacts with Smad3 to inhibit Smad3-mediated AP-1 transactivation[20]. AP-1 bound to *MYC* and *XIAP* promoter and activated their transcription[33,34]. In this study, SH3RF2 was found to promote AP-1 transactivation and *MYC* and *XIAP* expression by promoting the proteasomal degradation of RBPMS. AP-1 and its target genes, including *MYC* and *XIAP* have been implicated in DDP resistance. In OC cells, inhibition of the DNA binding activity of AP-1 mitigated DDP resistance[35]. Inhibition of c-Myc and XIAP re-sensitizes resistant OC cells to DDP[36,37]. We speculate that RBPMS may inhibit DDP resistance by indirectly suppressing *MYC* and *XIAP* transcription through AP-1 or by directly suppressing their transcription.

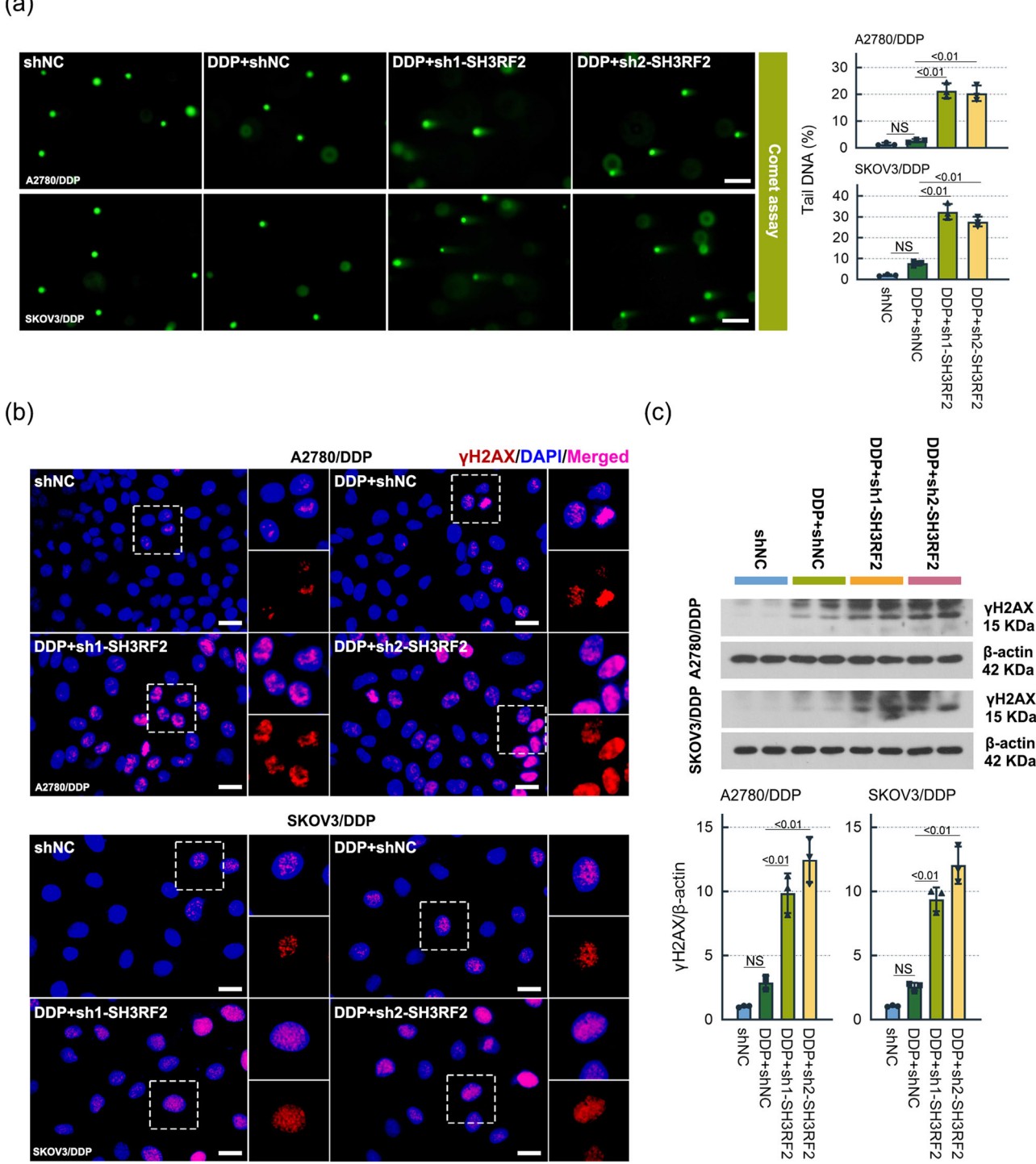

**Fig. 4 Effect of SH3RF2 depletion on DDP-induced DNA damage in OC in vitro.** DDP-resistant OC cells with or without depletion of SH3RF2 were exposed to DDP (A2780/DDP: 30 µM; SKOV3/DDP: 22 µM) for 48 h. $n = 3$. **a** Comet assays of DDP-resistant OC cells (Scale bars, 100 µm). $n = 3$. **b** Immunofluorescence of γH2AX in DDP-resistant OC cells (Scale bars, 25 µm). $n = 3$. **c** Western blots of lysates from DDP-resistant OC cells stained for γH2AX. $n = 3$. Data are expressed as the mean ± SD. The $p$ values were determined by one-way ANOVA. NS, no significance.

There are limitations in this study. We only chose two frequently used OC cell lines to investigate the mechanism of DDP resistance in vitro. The cell lines from other OC subtypes should also be used for the exploration of the mechanism of DDP resistance. The present study reports how SH3RF2 modulates DDP resistance. It is not verified whether RBPMS inhibits DDP resistance and *MYC* and *XIAP* transcription by suppressing AP-1 transactivation. It is necessary to define the molecular mechanisms underlying the modulation of DDP resistance by RBPMS in our subsequent work. In addition, we mainly focused on the effect of SH3RF2 on DDP-resistant OC cells. In our future work, we will explore whether SH3RF2 has an impact on the sensitivity of

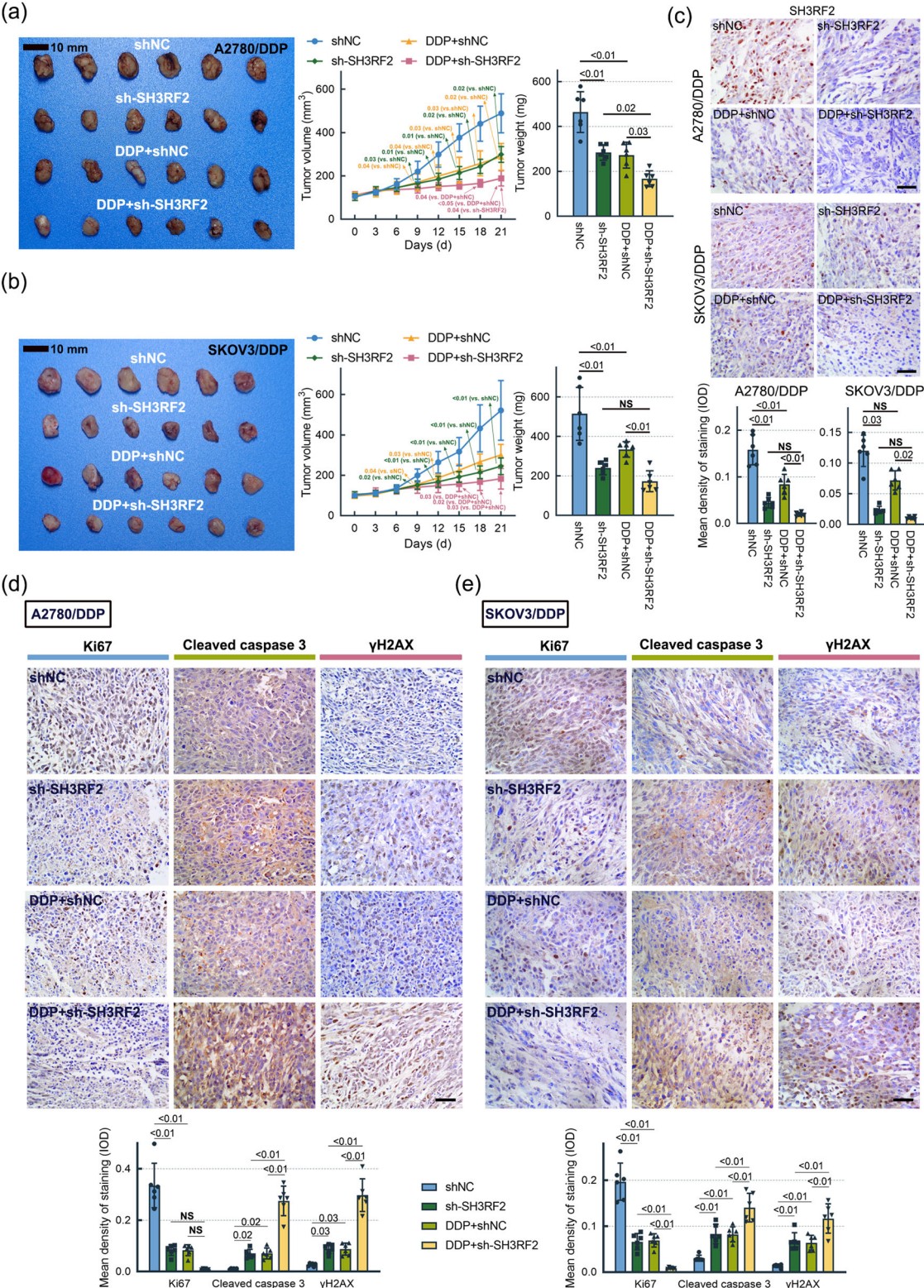

**Fig. 5 Effect of SH3RF2 depletion on DDP resistance in the OC xenograft model.** The OC xenograft model was established by subcutaneous injection of DDP-resistant OC cells ($1 \times 10^6$ cells per mice). **a**, **b** Tumor volumes were quantified every 3 days ($p < 0.01$ or $p < 0.05$ vs. the shNC/DDP+shNC/shSH3RF2 group) and the tumors were weighed 21 days after injection. $n = 6$. **c–e** IHC staining of SH3RF2, Ki67, cleaved caspase 3, and γH2AX in the tumor tissues (Scale bars, 50 μm). $n = 6$. Data are expressed as the mean ± SD. The $p$ values were determined by Kruskal-Wallis test or two-way ANOVA. NS, no significance.

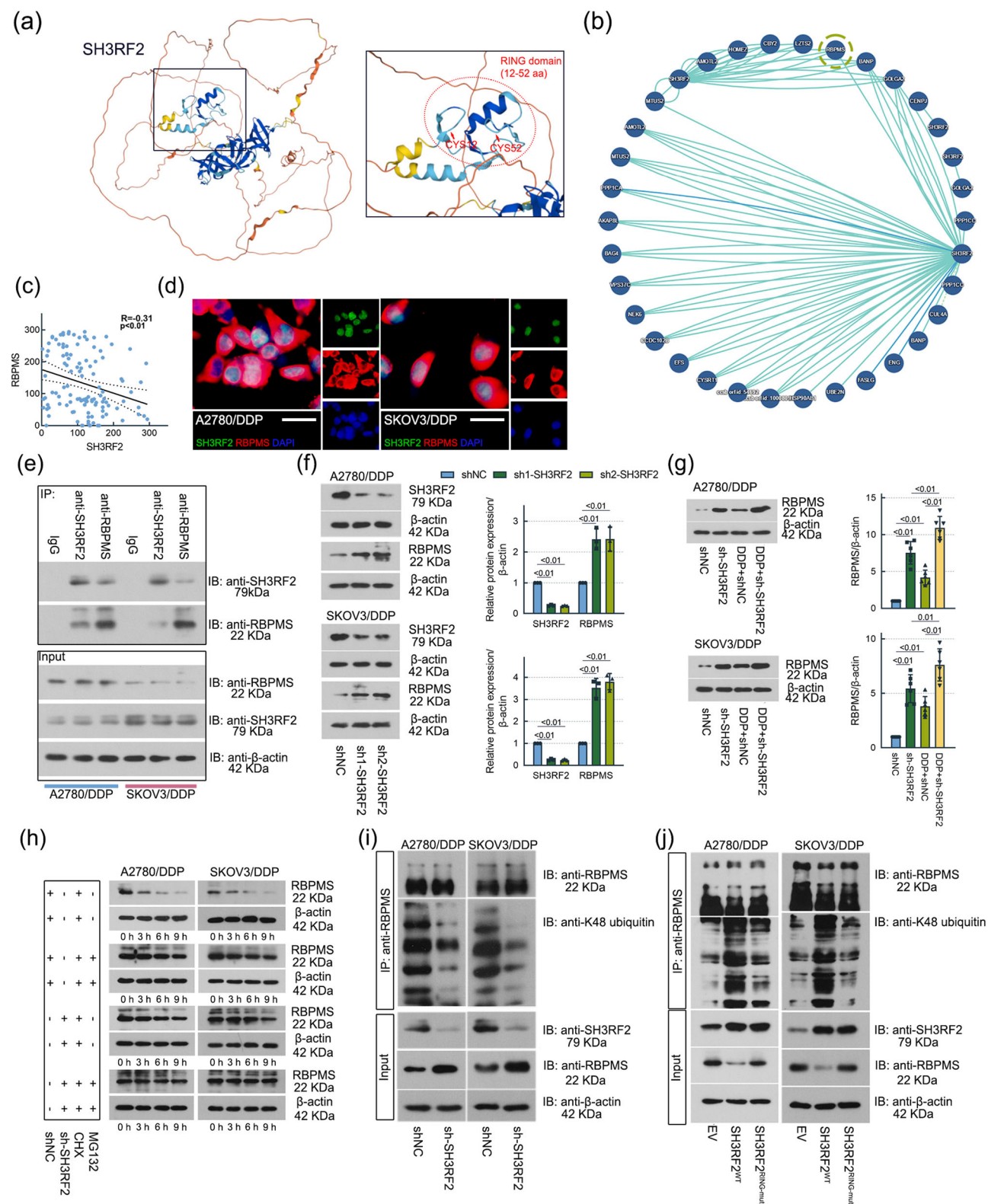

parental OC cells to DDP, which is useful to uncover the candidate target for overcoming of DDP-based chemotherapy toxicity.

In summary, the study demonstrates the biological significance of SH3RF2 in platinum resistance in OC. Increased SH3RF2 was related to acquired DDP resistance in OC by inducing proteasomal degradation of RBPMS. Targeting SH3RF2 may be a potential way to sensitize DDP-resistant OC. Our findings

provide insights into understanding the molecular mechanism associated with platinum resistance in OC.

## Materials and methods

**Patient samples.** A total of 120 formalin-fixed paraffin-embedded tumor samples, including 60 DDP-sensitive and 60 DDP-resistant

**Fig. 6 SH3RF2 impaired the stability of RBPMS via its E3 ubiquitin ligase activity. a** 3D structure of SH3RF2 from the UniProt database (https://www.uniprot.org/). SH3RF2 contains a RING domain (12-52 aa) and exerts E3 ubiquitin ligase activities. **b** The PPI network analysis from the IntAct Molecular Interaction Database (https://www.ebi.ac.uk/intact/home) exhibits that SH3RF2 interacts with 25 proteins including RBPMS. **c** Pearson correlation analysis between IHC scores of SH3RF2 and RBPMS in the tumor tissues from OC patients. $n = 120$. **d** Colocalization of endogenous SH3RF2 and RBPMS was detected by immunofluorescence (Scale bars, 25 μm). $n = 3$. **e** Co-immunoprecipitation of SH3RF2 with RBPMS using anti-SH3RF2 and anti-RBPMS. Immunoblotting was performed with anti-RBPMS and anti-SH3RF2. $n = 3$. **f** SH3RF2 and RBPMS expressions in DDP-resistant OC cells with stable depletion of SH3RF2 were evaluated by western blotting. $n = 3$. **g** RBPMS expressions in the xenograft tumor tissues generated by DDP-resistant OC cells with stable depletion of SH3RF2 were evaluated by western blotting. $n = 6$. **h** DDP-resistant OC cells with stable depletion of SH3RF2 were treated with 20 μg/mL CHX or/and 5 μM MG132 for 0, 3, 6, and 9 h. Western blots of lysates from cells stained for RBPMS. $n = 3$. **i** Co-immunoprecipitation of RBPMS with K48 ubiquitin in DDP-resistant OC cells with stable depletion of SH3RF2 using anti-RBPMS. Immunoblotting was performed with anti-RBPMS and anti-K48 ubiquitin. $n = 3$. **j** Co-immunoprecipitation of RBPMS with K48 ubiquitin in DDP-resistant OC cells transfected with wild-type SH3RF2 or the RING mutant of SH3RF2 using anti-RBPMS. Immunoblotting was performed with anti-RBPMS and anti-K48 ubiquitin. $n = 3$. Data are expressed as the mean ± SD. The $p$ values were determined by one-way or two-way ANOVA.

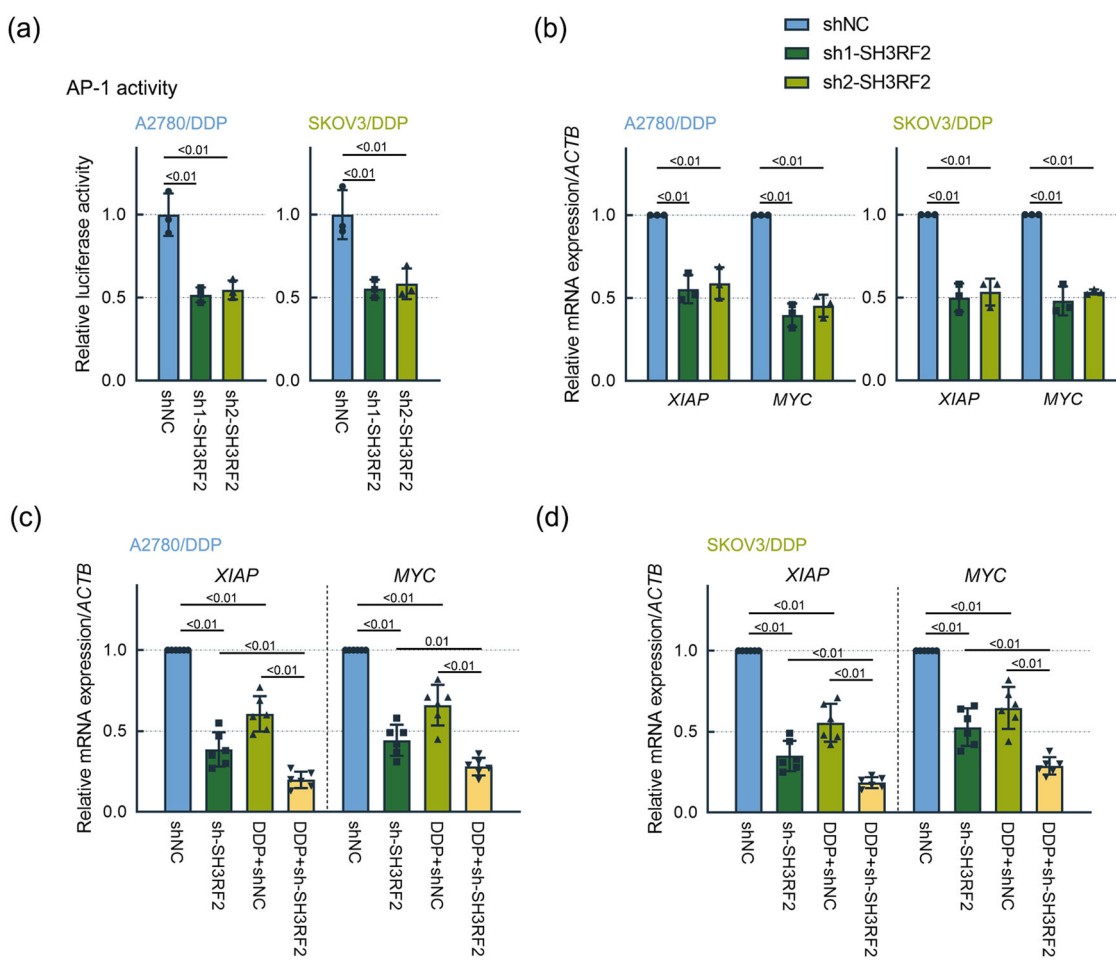

**Fig. 7 Effect of SH3RF2 depletion on AP-1 signaling in DDP-resistant OC cells. a** To determine the AP-1-dependent transcriptional activities, DDP-resistant OC cells were co-transfected with pAP1-Ta-luc and pRL-TK plasmids. Luciferase activity assays were performed at 48 h after cell transfection. $n = 3$. **b** RT-qPCR analysis of AP-1 downstream genes *XIAP* and *MYC* in DDP-resistant OC cells with stable depletion of SH3RF2. $n = 3$. **c, d** RT-qPCR analysis of AP-1 downstream genes *XIAP* and *MYC* in the xenograft tumor tissues generated by DDP-resistant OC cells with stable depletion of SH3RF2. $n = 6$. Data are expressed as the mean ± SD. The $p$ values were determined by one-way or two-way ANOVA.

samples, were obtained from patients with OC who received DDP-based chemotherapy at the Shengjing Hospital of China Medical University between 2019 and 2022. Patients with tumors that progress during 6 months or recurrence within 6 months after DDP treatment were DDP-resistant, while those with tumor recurrence greater than 6 months after DDP treatment were DDP-sensitive. Informed consent was obtained from all participants and this study was approved by the Ethics Committee of Shengjing Hospital of China Medical University, and all

procedures were conducted in accordance with the Declaration of Helsinki.

**Human cell lines and cell culture**. Two human OC cell lines including A2780 and SKOV3 were purchased from iCell Bioscience Inc (China). Two cell lines used in the study were authenticated by STR analysis. Mycoplasma contamination was also tested. The A2780 cells were cultured in RPMI-1640 medium (Solarbio, China)

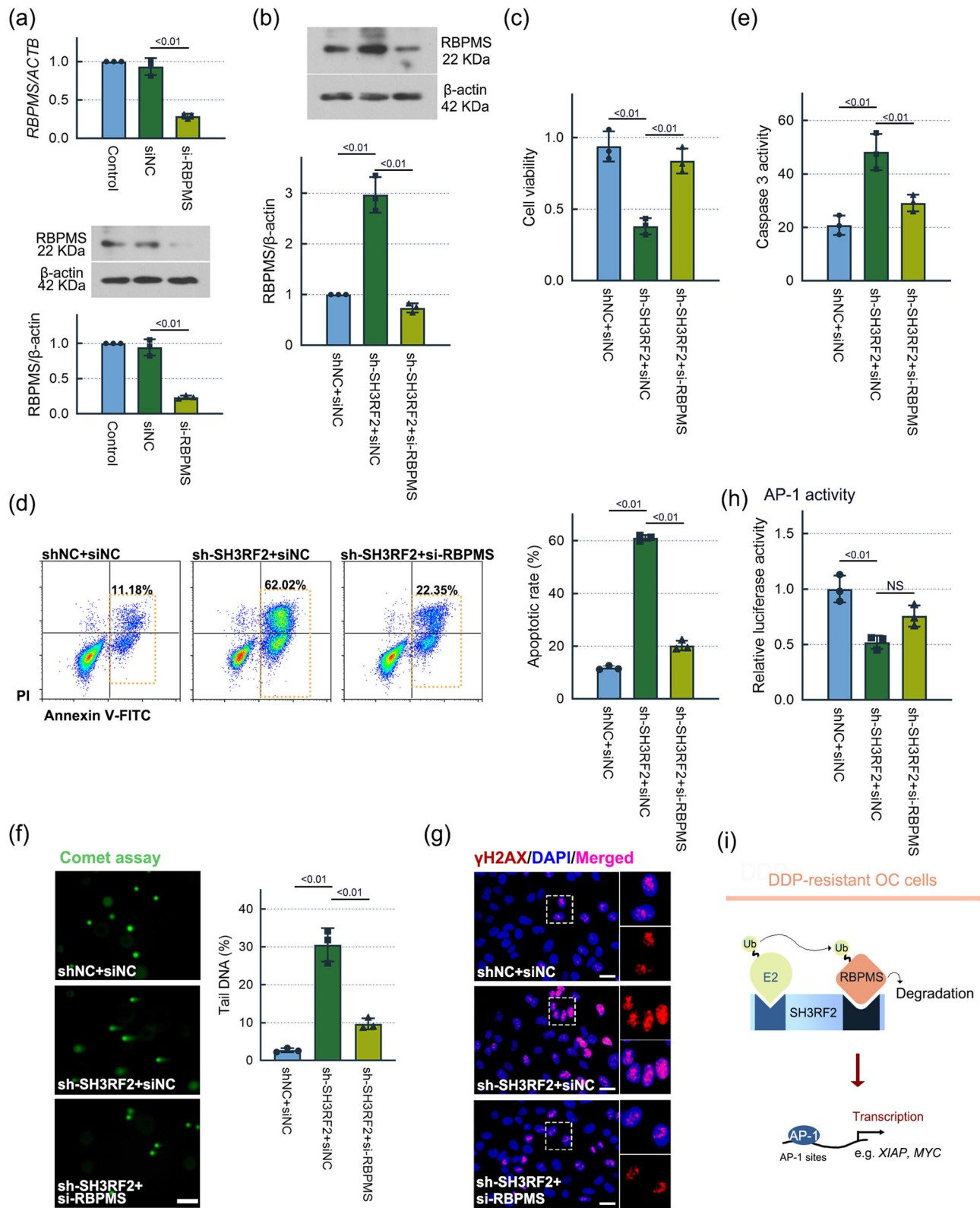

with 10% fetal bovine serum (FBS, TIANHANG, China) and the SKOV3 cells were cultured in McCoy's 5 A medium (Servicebio, China) with 10% FBS at 37 °C with 5% $CO_2$.

**Screening of platinum-resistant cells**. The DDP-resistant cell lines A2780/DDP and SKOV3/DDP cells were obtained through exposing cells to increasing concentrations of DDP (2, 4, 8, 16, 32, and 64 μM; MeilunBio, China) for 6 cycles (4-hour DDP treatment, followed by 3-week regular cell culture without DDP treatment)[38]. The CP-resistant cell lines A2780/CP and SKOV3/CP cells were obtained through exposing cells to increasing concentrations of CP (10-100 μg/mL; MeilunBio).

**Fig. 8 SH3RF2 contributed to DDP resistance in OC cells by inhibition of RBPMS. a** DDP-resistant A2780 cells were transfected with RBPMS siRNA and RBPMS expressions were evaluated by RT-qPCR ($n = 3$) and western blotting ($n = 3$) at 24 h after cell transfection. DDP-resistant A2780 cells with depletion of SH3RF2 were transfected with RBPMS siRNA for 24 h and exposed to 30 μM DDP. **b** Western blots of lysates from cells stained for RBPMS at 48 h after DDP treatment. $n = 3$. **c** MTT assays at 48 h after DDP treatment. $n = 3$. **d** Cell apoptosis were determined by flow cytometry following Annexin V-FITC/PI double staining at 48 h after DDP treatment. $n = 3$. **e** Caspase 3 activities at 48 h after DDP treatment. $n = 3$. **f** Comet assays at 48 h after DDP treatment. $n = 3$. **g** Immunofluorescence of γH2AX at 48 h after DDP treatment (Scale bars, 25 μm). $n = 3$. **h** DDP-resistant A2780 cells with depletion of SH3RF2 were co-transfected with RBPMS siRNA, pAP1-Ta-luc plasmids, and pRL-TK plasmids for 24 h and exposed to 30 μM DDP. Luciferase activity assays were performed at 48 h after DDP treatment. $n = 3$. **i** Schema of the potential mechanism of DDP resistance in OC cells. NS, no significance. Data are expressed as the mean ± SD. The $p$ values were determined by one-way ANOVA.

**Screening of stable cell lines.** A2780/DDP and SKOV3/DDP cells were infected with lentiviral vectors containing short hairpin RNA (shRNA) targeting SH3RF2 (sh-SH3RF2) or its negative control non-targeting shRNA (shNC). Forty-eight hours post-infection, the cells were selected with 1 μg/mL purinomycin (Solarbio) for 1 week. The sh-SH3RF2 or shNC was inserted into the pLVX-shRNA1 vector (FENGHUISHENGWU, China) between *Bam*HI and *Eco*RI and their target sequences were listed as follows: sh1-SH3RF2, 5′-GCTCCGTGGAAGTCATCAAGC-3′; sh2-SH3RF2, 5′-GCTAGTGCCTAATGTCAGAAT-3′; shNC, 5′-TTCTCCGAACGTGTCACGT-3′.

**Cell transfection.** A2780/DDP cells with stable knockdown of SH3RF2 were transiently transfected with RBPMS siRNA or negative control siRNA (siNC) by Lipofectamine 3000 (Invitrogen, USA) following manufacturer's instructions. A2780/DDP cells were transiently transfected with empty pcDNA3.1 vectors or pcDNA3.1 vectors expressing wild-type SH3RF2 or the RING mutant of SH3RF2 by Lipofectamine 3000 (Invitrogen). RBPMS siRNA (sense, 5′-GGCUAUGAGGGUUCUCUUAUU-3′; anti-sense, 5′-UAAGAGAACCCUCAUAGCCUU-3′), wild-type SH3RF2, and the RING mutant of SH3RF2 were synthesized by GENERAL BIOL (China).

**MTT assay.** A2780/DDP and SKOV3/DDP cells were exposed to 0, 2.5, 5, 10, 20, 40, or 80 μM DDP for 48 h. A2780/CP and SKOV3/CP cells were exposed to 0, 1, 10, 100, or 1000 μM CP for 48 h. A2780/DDP cells with stable knockdown of SH3RF2 were transiently transfected with RBPMS siRNA for 24 h and then treated with 30 μM DDP for 48 h. The MTT assays were performed according to manufacturer's instructions. In brief, the MTT (50 μL; KeyGEN, China) was added to each well. After 4-h incubation with MTT, the supernatants of cell cultures were removed and then dimethyl sulfoxide (DMSO; 150 μL; KeyGEN) was added. The optical density (OD) values were measured at 490 nm by the BioTek® 800™ TS Absorbance Reader (BioTeK, USA).

**Reverse transcription-quantitative PCR (RT-qPCR).** RNA was isolated from cells using TRIpure (BioTeke Corporation, China) and its concentration was measured by the NanoDrop™ 2000 Spectrophotometer (Thermofisher, USA). RNA was converted to cDNA by performing reverse transcription with the BeyoRT II M-MLV reverse transcriptase (Beyotime, China). Quantitative real-time PCR was performed on the Exicycler™ 96 PCR system (Bioneer, China) using SYBR Green (Solarbio) following the manufacturer's manual. Primer sequences are available in Table 2. All data were normalized to *ACTB*.

**Western blotting.** Cultured cells and tumor tissues were lysed in RIPA Lysis Buffer (Beyotime) supplemented with phenylmetha-nesulfonyl fluoride (PMSF, Beyotime) to extract total protein. Nuclear protein was extracted using the Nuclear and Cytoplasmic Protein Extraction Kit (Beyotime) according to the manu-facturer's instructions. Protein samples from each group were loaded on a 10% or 14% SDS-polyacrylamide gels (SDS-PAGE) and transferred to polyvinylidene difluoride (PVDF) membranes (Thermofisher), followed by blocking with 5% bovine serum albumin (BSA; Biosharp, China) for 1 h. Immunodetection was conducted with antibodies against SH3RF2 (1:300, Santa cruz, USA, #sc-100976), cleaved poly ADP-ribose polymerase (PARP; 1:500, Affinity, China, #AF7023), cleaved caspase 3 (1:1000, Affinity, #AF7022), phosphorylated histone H2AX (γH2AX; 1:500, Affinity, #AF3187), RNA binding protein with multiple splicing (RBPMS; 1:1000, Proteintech, USA, #15187-1-AP), β-actin (1:20000, Proteintech, #66009-1-Ig), and Histone H3 (1:1000, Proteintech, #17168-1-AP). Secondary antibodies including horseradish peroxidase-conjugated goat antirabbit IgG and goat anti-mouse IgG (Proteintech, #SA00001-1 or #SA00001-2) were used at 1:10000 dilution. Protein bands were visualized using ECL reagent. Expression data were normalized to β-actin or Histone H3.

**Flow cytometry assay.** Cell apoptosis detection was performed using the Annexin V-FITC/PI Apoptosis Detection Kit (Key-GEN) according to the manufacturer's manual. In brief, cells were resuspended in 500 μL Binding Buffer, stained with Annexin V-FITC and Propidium Iodide (PI) for 15 min in the dark, and analyzed using the NovoExpress software (version 1.4.1, Agilent, USA) with the NovoCyte Flow Cytometer System (Agilent). Gating strategy was shown in Supplementary Fig. 6.

**AO/EB staining.** Cells were fixed 4% Polyformaldehyde (PFA) for 15 min and washed with PBS. The cells were stained with AO/EB staining solution (Yuanye Bio-Technology, China) according to the manufacturer's instructions. The cells were examined under a fluorescent microscope (OLYMPUS, Japan).

**Caspase 3 activity assay.** Activities of caspase 3 were measured using the caspase 3 activity detection kits (Beyotime) following the supplier's recommendations. In brief, Ac-DEVD-*p*NA was used as the substrate and the activity of caspase 3 was analyzed by quantifying the enzyme-catalyzed release of *p*NA at 405 nm using the ELX800 Microplate reader (BioTek).

| Table 2 Primer sequences for RT-qPCR. | |
| --- | --- |
| | **Sequences (5′-3′)** |
| *SH3RF2* Forward | CGTGGTGGTGGAGATGG |
| *SH3RF2* Reverse | TGGGAGGTGTAATGTTTGGTG |
| *RBPMS* Forward | CAAGAACAAACTCGTAGGGACT |
| *RBPMS* Reverse | GCGGGATAGGTGAAAGC |
| *XIAP* Forward | CACAGGCGACACTTTCC |
| *XIAP* Reverse | TTAGCCCTCCTCCACAG |
| *MYC* Forward | ACACCCTTCTCCCTTCG |
| *MYC* Reverse | CCGCTCCACATACAGTCC |

**Colony formation assay**. Cells (300 cells per plate) were plated onto a culture plate and cultured in DDP (A2780/DDP cells: 30 μM; SKOV3/DDP cells: 22 μM). After 2 weeks, the visible colonies were stained with the Wright's-Giemsa reagent (Key-GEN) following fixation with 4% PFA for 25 min at room temperature. The colonies that consist of at least 50 cells were counted and the colony formation rate was calculated as: formed colonies / seeded cells × 100%.

**Sphere formation assay**. Cells ($5 \times 10^4$ cells per well) were plated on ultra-low attachment plates and cultured using the Mammo-Cult™ Human Medium Kit (STEMCELL, Canada). After 10 days, tumor spheres with the diameter of >75 μm were quantified.

**Comet assay**. The slides were coated with 0.5% normal melting point agarose and then cells were mixed with 2% low melting point agarose and placed onto the slides. After solidification of agarose at 4°C, the slides were immersed in lysis solution overnight at 4°C. The slides were incubated in an electrophoresis chamber with pre-cool electrophoresis buffer for 45 min and electrophoresis was performed at 35 volts for 35 min. After staining with GoldView, DNA damage was observed under a fluorescence microscope (OLYMPUS, Japan).

**Immunofluorescence staining**. Cells ($2 \times 10^5$ cells per well) were cultured on the cell climbing films. The prepared cell climbing films were fixed by 4% PFA (Sinopharm, China) for 15 min, permeabilized with 0.1% TritonX-100 (Beyotime) for 30 min at room temperature, and pre-incubated in 1% BSA (Sangon, China) for 15 min at room temperature. For γH2AX staining, the cell climbing films were incubated overnight at 4°C with antibodies against γH2AX (1:50, Thermofisher, #PA5-97354) and then incubated for 60 min at room temperature with Alexa Fluor™ 555-labeled goat anti-rabbit IgG (1:200; Invitrogen, #A27039; Red). For SH3RF2-RBPMS staining, the cell climbing films were incubated overnight at 4°C with antibodies against SH3RF2 (1:100, Thermofisher, #PA5-63527) and RBPMS (1:50, Santa cruz, #sc-293285) and then incubated for 60 min at room temperature with FITC-labeled goat anti-rabbit IgG (1:200; Abcam, UK, #ab6717; Green) and Alexa Fluor™ 555-labeled goat anti-mouse IgG (1:200; Invitrogen, #A-21424; Red). The cell nuclei were counterstained with DAPI (Aladdin, China). The images were captured with a fluorescence microscope (Olympus, Japan).

**Subcutaneous xenograft model**. Animal experiments were approved by the Ethics Committee of Shengjing Hospital of China Medical University, and all procedures were performed in compliance with the *Guide for the Care and Use of Laboratory Animals*.

For xenograft experiments, stable cell lines ($1 \times 10^6$ cells per mouse) were injected subcutaneously into 7-week-old female BALB/c nude mice (Huachuang Sino, China). When the tumors reached ~100 mm³ in volume, the mice were intraperitoneally injected with 5 mg/kg body weight DDP twice a week. The tumor size was measured every 3 days using a caliper, and the volume of tumor was calculated. The tumors were excised at 21 d after the first DDP injection and weighed. Tumor tissues were harvested to perform western blot analysis, RT-qPCR analysis, or IHC staining. The tumor tissues used for IHC staining were fixed in 10% formalin and embedded in paraffin.

**IHC staining**. The paraffin-embedded tumor tissues from mice or OC patients were cut in 5-μm-thick sections. The sections were deparaffinized in xylene, rehydrated through decreasing ethanol series ending in distilled water, and boiled in antigen retrieval solution in a microwave oven at low power for 10 min. Then, the sections were incubated in 3% $H_2O_2$ for 15 min at room temperature to block endogenous peroxidase activities, followed by blocking of non-specific antigens in 1% BSA for 15 min at room temperature. The tissue sections were incubated overnight at 4°C with antibodies against SH3RF2 (1:50, Thermofisher, #A5-63527), RBPMS (1:50, Santa Cruz, #sc-293285), Ki67 (1:50, Affinity, China, #AF0198), cleaved caspase 3 (1:50, Affinity, #AF7022), and γH2AX (1:50, Thermofisher, #PA5-97354) and then incubated for 60 min at 37°C with horseradish peroxidase-conjugated goat antirabbit IgG or goat anti-mouse IgG (1:500; Thermofisher, #31460 or #31430). Incubation with DAB substrate (MXB Biotechnologies, China) and counterstaining with hematoxylin (Solarbio) were carried out. The images were acquired with a microscope (Olympus, Japan). SH3RF2 and RBPMS staining in tumor tissues were quantified by two pathologists. The IHC scores were calculated using the following formula: the percentage of positive cells (0–100%) multiplied by its staining intensity (0 = negative, 1 = weak, 2 = moderate, 3 = strong). The median value of IHC scores was chosen as the cutoff point to separate the tumor tissues with high SH3RF2 expression (IHC score > median) from those with low SH3RF2 expression (IHC score ≤ median). The mean intensity of SH3RF2, Ki67, cleaved caspase 3, and γH2AX staining was calculated by dividing the integrated optical density (IOD) by the area.

**Co-immunoprecipitation assay**. Cultured cells were lysed in Cell lysis buffer for Western and IP (Beyotime) supplemented with PMSF, and the supernatants were collected after centrifugation at 10,000 × g for 5 min at 4°C. For the co-immunoprecipitation of endogenous proteins, anti-SH3RF2 (Santa cruz, #sc-100976) or anti-RBPMS (Proteintech, #15187-1-AP) antibodies immobilized with AminoLink plus coupling resin for antibody immobilization at room temperature for 2 h using the Pierce™ Co-Immunoprecipitation Kit (Thermofisher) as per the manufacturer's protocols and then the total protein lysates were added to AminoLink plus coupling resin. The immunoprecipitates were eluted with elution buffer and the eluates were analyzed by western blot detection with primary antibodies and secondary antibodies as described above. Primary antibodies were listed as follows: antibodies against SH3RF2 (1:300, Santa cruz, #sc-100976), RBPMS (1:1000, Proteintech, #15187-1-AP), K48 ubiquitin (1:500, Abclonal, China, #A18163), and β-actin (1:20000, Proteintech, #66009-1-Ig). Secondary antibodies including horseradish peroxidase-conjugated goat antirabbit IgG and goat anti-mouse IgG (Proteintech, #SA00001-1 or #SA00001-2) were used at 1:10,000 dilution.

**Detection of protein stability**. Cells were treated with 20 μg/mL CHX to inhibit protein synthesis or/and 5 μM MG132 to block the proteolytic activity of the 26 S proteasome complex. The cells were harvested at 0, 3, 6, or 9 h after CHX/MG132 treatment. The levels of RBPMS were detected by western blotting using antibody against RBPMS (1:1000, Proteintech, #15187-1-AP).

**Dual-luciferase reporter assay**. Stable cell lines with depletion of SH3RF2 were transfected with 1.5 μg pAP1-TA-luc reporter plasmids (Beyotime) and 1.5 μg pRL-TK plasmids (Promega, USA) Lipofectamine 3000 (Invitrogen, USA) following manufacturer's instructions. One microgram pAP1-TA-luc reporter plasmids and 1 μg pRL-TK plasmids were transfected with 30 pmol RBPMS siRNA into stable cell lines with depletion of SH3RF2 by Lipofectamine 3000. The relative activity of luciferase was measured using the Dual Luciferase Reporter Gene Assay Kit

(KeyGEN) as per the manufacturer's protocols. The firefly luciferase activity was normalized to the renilla luciferase activity.

**Statistics and reproducibility.** Statistical analysis was performed using the GraphPad Prism software (version 8.0, GraphPad Prism Software Inc., USA). When data were not normally distributed, non-parametric tests (Kruskal-Wallis test or Mann-Whitney U test) were used. When data were normally distributed, the unpaired Student's t-test was used for statistical comparison of two groups. Welch's t-test was used when the variances between two groups were significantly different. For statistical comparison of three or more groups, the one-way or two-way analysis of variance (ANOVA) followed by Tukey's or Sidak post hoc test was used. The correlation between SH3RF2 expression levels and clinicopathological parameters of OC patients was analyzed by Chi-square test or Chi-square test with Yates' correction. The $p$ values < 0.05 were considered statistically significant. Each experiment was performed with three or six independent biologic replicates. Data are expressed as mean and standard deviation (mean ± SD).

**Reporting summary.** Further information on research design is available in the Nature Portfolio Reporting Summary linked to this article.

## Data availability

All uncropped images of western blot analysis are available in Supplementary Fig. 7. The source data are provided in Supplementary Data. The authors declare that all other data used to support the findings in the present study will be available to all readers.

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

## Acknowledgements

This work was supported by the National Key R&D Program of China (No. 2022YFC2704200 and 2022YFC2704203 to Q-JW), National Natural Science Foundation of China (No. 82073647 and No. 82373674 to Q-JW, and No. 82103914 to T-TG),

Outstanding Scientific Fund of Shengjing Hospital (No. M1150 to Q-JW), 345 Talent Project of Shengjing Hospital of China Medical University (No. M0952 to T-TG).

## Author contributions

T.-T.G., F.-H.L., Q.X. and Q.-J.W. conceived the study and contributed to the design. T.-T.G., F.-H.L., Q.X., L.W. and Q.-J.W. wrote the original draft. F.-H.L., Q.X., F.-L.J., T.T., Q.-P.M., X.Q., Y.S. and S.G. collected the data. Y.-Z.L., Y.-F.W., H.-L.X., F.C. and M.-L.S. cleaned and analyzed the data. All authors interpreted the data, read the manuscript, and approved the final vision. T.-T.G., F.-H.L., Q.X., X.Q., Y.-H.Z., D.-H.H. and Q.-J.W. contributed to the revision. T.-T.G., F.-H.L. and Q.X. contributed equally to this work.

## Competing interests

The authors declare no competing interests
