## [Peer Review File · Communications Biology]

Reviewers' comments:

Reviewer #1 (Remarks to the Author):

Gong et al reported that SH3RF2 was elevated in cisplatin-resistant ovarian cancer. They proposed that SH3RF2, an E3 ligase, promoted ubiquitination of RNA-binding protein RBPMS to increase AP-1 transactivation in ovarian cancer cells. SH3RF2 depletion led to apoptosis and DNA damage in cisplatin treated cells. Although this manuscript provides some interesting cisplatin-resistant mechanisms, the data quality and results provided at this stage are not sufficient to claim that the SH3RF2-RBPMS-AP1 axis is a key mechanism in ovarian cancer.

1. Cell lines: A2780 is an endometrioid cancer cell line and SKOV3 is a serous cystadenocarcinoma. Can these cell lines sufficiently represent each ovarian cancer subtype to demonstrate that SH3RF2-RBPMS-AP1 is a common cisplatin resistance mechanism?
2. Western blots: All western blots need to have proper labels of molecular weight markers.
3. Comet assays: For all comet assays in the manuscript, only one cell per condition was shown. The authors need to show that the same phenomenon can be observed in majority of the cells. The authors also should try to quantify and compare the difference to the control group.
4. rH2AX IF staining quality: The pattern of rH2AX is really strange. No foci can be seen. The authors should also show enlarged image with high resolution for this data.
5. Fig. 4 IHC. The authors should also show the SH3RF2 level in the xenografted tumors.
6. Fig. 5 SH3RF2-RBPMS interaction. Fig. 5d shows weak interaction between SH3RF2 and RBPMS in SKOV3-DDP cells. This raises a question whether the interaction can only be seen upon cisplatin treatment. Since endogenous SH3RF2 and RBPMS can be detected in parental cells, the authors should also use the parental SKOV3 to perform the Co-IP side-by-side with the SKOV3-DDP cells to demonstrate that SKOV3-DDP indeed has increased interaction between these two proteins.
7. Fig. 5g. If RBPMS is ubiquitinated by SH3RF2, under MG132 treatment, the blot of RBPMS should show multiple bands. The authors need to show the uncut blot image.
8. Line 156: What did the authors mean by "RBPMS binds to AP-1 members"?
9. For the data presented now, there is no solid evidence to show that XIAP and MYC downregulation is through AP-1 suppression by RBPMS. It is also possible that RBPMS directly suppressed XIAP and MYC.
10. Line 157: "SH2RF2" should be "SH3RF2".

Reviewer #2 (Remarks to the Author):

In this manuscript Gong et al. identify the E3 ligase SH3RF2 as a marker of cisplatin resistance in ovarian cancer. The authors go on to describe SH3RF2 mediated ubiquitination of RBPMs as the potential mechanism behind this effect. Overall, this is a very well written paper and the experiments are performed in a methodical and logical manner. However, a few minor corrections and experiments are suggested to enhance the overall quality of the paper.

Minor issues.

1. For the benefit of the reader the authors must define the difference between ddp-resistant and cp-resistant.
2. Page line 94. "MTT assays exhibited..." The sentence is poorly written and is grammatically incorrect.
3. Page 6 line 94..to analyze.
4. The authors need to explain why they randomly looked at SH3RF2.
5. Figure 2, 2b, 2e and others like it where the authors attempt to show both cell lines in the same graph. Up down or side to side. I understand the figure but it is suggested that the authors make to separate graphs for each of the cell lines. It makes it easier for the reader to understand.
6. Figure 2c cell lines have not been annotated in the figure.
7. Figure 2E. Is the labelling of the figure correct. CC3 tends to give two bands 13 and 17 kDa and cleaved parp gives one band at 90 kDa. This is not what is shown in the figure.
8. Although this paper attempts to demonstrate that SH3RF2 is a critical factor in regulating cisplatin resistance. It is important that the authors demonstrate that in WT cells that SH3RF2 knockdown increases sensitivity of these cells to low levels of cisplatin. This may be a major argument to overcome chemotherapy toxicity if SH3RF3 therapy is ever implemented.
9. In figure 5E sh3rf2 knockdown must be demonstrated.
10. In figure 5G RBPMs must be shown in the whole cell lysate. It is recommended the authors do this same experiment with overexpression of SH3RF2 and a catalytically inactive mutant of SH3RF2 to determine that the catalytic effect of SH3RF2 is critical for these effects and that SH3RF2 is not acting as a scaffold protein.
11. Figure 5f. Must be better explained. The reviewer does not understand anything in this figure. What does X mean? What does the circle mean? If X means knockdown shouldn't there be an increase in RBPMs. Should the same effect be seen in blot 3 as in blot 1 as this is the same experiment.

Reviewer #3 (Remarks to the Author):

In 'SH3RF2 contributes to cisplatin resistance in ovarian cancer cells by promoting RBPM2 degradation', the author innovatively identified the role of SH3RF2, an E3 ligase, in ovarian cancer (OC) cisplatin (DDP) resistance and explored the mechanisms it involved.

The authors have successfully established DDP-resistant cell lines based on A2780 and SKOV3 and reveal a significant increase of SH3RF2 in DDP-resistant cell line. Then knocked down SH3RF2 in the DDP-resistant cell lines was performed and demonstrated the loss of DDP-resistance in SH3RF2 knockdown cell lines. Through PPI network analysis, RBPM2 was identified. The effect of SH3RF2 in mediating K48-linked ubiquitination of RBPM2 was demonstrated.

Major comments:

1. Cisplatin resistance is often related to cancer stemness. Have any change in stem-like properties or markers been observed.
2. Do RBPM2 and SH3RF2 colocalise? It seems RBPM2 is cytoplasmic whereas SH3RF2 is nuclear (Fig. 5c). Where are they colocalized?
3. A2780 has a paired drug resistant line. Has similar observation been seen using this paired cell line. It will be a more stringent test.
4. The SH3RF2-RBPM2-AP-1 axis is not thoroughly tested in mice.
5. Is there an association of SH3RF2-RBPM2-AP-1 axis with patients' samples?

Reviewers' comments:

Reviewer #1:

Gong et al reported that SH3RF2 was elevated in cisplatin-resistant ovarian cancer. They proposed that SH3RF2, an E3 ligase, promoted ubiquitination of RNA-binding protein RBPMS to increase AP-1 transactivation in ovarian cancer cells. SH3RF2 depletion led to apoptosis and DNA damage in cisplatin treated cells. Although this manuscript provides some interesting cisplatin-resistant mechanisms, the data quality and results provided at this stage are not sufficient to claim that the SH3RF2-RBPMS-AP1 axis is a key mechanism in ovarian cancer.

1. Cell lines: A2780 is an endometrioid cancer cell line and SKOV3 is a serous cystadenocarcinoma. Can these cell lines sufficiently represent each ovarian cancer subtype to demonstrate that SH3RF2-RBPMS-AP1 is a common cisplatin resistance mechanism?

>>Reply

Serous cystadenocarcinoma and endometrioid carcinomas are considered the two most common malignant ovarian neoplasm. It was a limitation in our study that we only chose two cell lines from the two most common malignant ovarian neoplasm to investigate the mechanism of SH3RF2 in cisplatin resistance *in vitro*. It needs to be further explored in our subsequent work whether the mechanism of SH3RF2 in cisplatin resistance in cell lines from other ovarian cancer subtypes is same as in cell lines from endometrioid carcinomas and serous cystadenocarcinoma. The limitation has been mentioned in the discussion section [the last paragraph].

2. Western blots: All western blots need to have proper labels of molecular weight markers.

>>Reply

We labelled the molecular weight of proteins in all western blot images.

3. Comet assays: For all comet assays in the manuscript, only one cell per condition was shown. The authors need to show that the same phenomenon can be observed in majority of the cells. The authors also should try to quantify and compare the difference to the control group.

>>Reply

We performed comet assays again according to your comments, and the percentage of tail DNA was quantified [Fig.4&8].

4. rH2AX IF staining quality: The pattern of rH2AX is really strange. No foci can be seen. The authors should also show enlarged image with high resolution for this data.

>>Reply

We performed γ H2AX staining again according to your comments and provided images with higher resolution (600 \times) [Fig.4&8].

5. Fig. 4 IHC. The authors should also show the SH3RF2 level in the xenografted tumors.

>>Reply

According to your comments, we provided IHC images for SH3RF2 to show the SH3RF2 level in the xenografted tumors [Fig.5c].

6. Fig. 5 SH3RF2-RBPMS interaction. Fig. 5d shows weak interaction between SH3RF2 and RBPMS in SKOV3-DDP cells. This raises a question whether the interaction can only be seen upon cisplatin treatment. Since endogenous SH3RF2 and RBPMS can be detected in parental cells, the authors should also use the parental SKOV3 to perform the Co-IP side-by-side with the SKOV3-DDP cells to demonstrate that SKOV3-DDP indeed has increased interaction between these two proteins.

>>Reply

According to your comments, we compared the interaction between SH3RF2 and RBPMS in parental SKOV3 and SKOV3-DDP cells [Fig.S4]. There was a stronger interaction in SKOV3-DDP cells.

7. Fig. 5g. If RBPMS is ubiquitinated by SH3RF2, under MG132 treatment, the blot of RBPMS should show multiple bands. The authors need to show the uncut blot image.

>>Reply

According to your comments, we re-provided the blot of RBPMS [Fig.6j].

8. Line 156: What did the authors mean by “RBPMS binds to AP-1 members”?

>>Reply

We modified the sentence to make it easy to understand. The sentence was revised to “RBPMS binds to the components of AP-1 including Fos, FosB, and FosL1, ...” [Page 7].

9. For the data presented now, there is no solid evidence to show that XIAP and MYC downregulation is through AP-1 suppression by RBPMS. It is also possible that RBPMS directly suppressed XIAP and MYC.

>>Reply

Emerging evidence has suggested that AP-1 was involved in the regulation of *XIAP* and *MYC* mRNA expression (doi: [org/10.1210/me.2011-1037;10.18632/oncotarget.23897](https://doi.org/10.1210/me.2011-1037;10.18632/oncotarget.23897)). It was indicated that *XIAP* and *MYC* were AP-1 target genes. To verify the transactivation of AP-1, we detected their expression in DDP-resistant ovarian cancer cell lines. We share your view that RBPMS may directly suppress XIAP and MYC and we have added the description in the discussion section [the fourth paragraph].

10. Line 157: “SH2RF2” should be “SH3RF2”.

>>Reply

“SH2RF2” was revised to “SH3RF2”.

Reviewer #2 (Remarks to the Author):

In this manuscript Gong et al. identify the E3 ligase SH3RF2 as a marker of cisplatin resistance in ovarian cancer. The authors go on to describe SH3RF2 mediated ubiquitination of RBPMS as the potential mechanism behind this effect. Overall, this is a very well written paper and the experiments are performed in a methodical and logical manner. However, a few minor corrections and experiments are suggested to enhance the overall quality of the paper.

Minor issues.

1. For the benefit of the reader the authors must define the difference between ddp-resistant and cp-resistant.

>>Reply

We defined the difference between ddp-resistant and cp-resistant [Page 4: The level of SH3RF2 was higher in cisplatin (DDP)-resistant OC patients and in platinum-resistant OC cells; Page 5: Subsequently, carboplatin (CP)-resistant A2780 and SKOV3 cells were used to evaluate the expression of SH3RF2].

2. Page line 94. "MTT assays exhibited..." The sentence is poorly written and is grammatically incorrect.

>>Reply

We have modified the sentence [Page 4-5: The results of MTT assays showed that the IC50 value of DDP in DDP-resistant A2780 and SKOV3 cells was higher, when compared with their parental cells].

3. Page 6 line 94..to analyze.

>>Reply

We have corrected it [Page 7].

4. The authors need to explain why they randomly looked at SH3RF2.

>>Reply

Our lab mainly focused on the function of SH3RF family members in cancers due to its oncogenic function. A GEO dataset GSE15709 revealed that SH3RF2 was higher in DDP-resistant A2780 than in DDP-sensitive A2780. Therefore, we investigated the effect of SH3RF2 on DDP resistance.

5. Figure 2, 2b, 2e and others like it where the authors attempt to show both cell lines in the same graph. Up down or side to side. I understand the figure but it is suggested that the authors make to separate graphs for each of the cell lines. It makes it easier for the reader to understand.

>>Reply

We separated graphs for each of the cell lines [Fig.3a, 3b, 3e, 4c, 6f & 7a].

6. Figure 2c cell lines have not been annotated in the figure.

>>Reply

We annotated the cell lines in Fig.3c.

7. Figure 2E. Is the labelling of the figure correct. CC3 tends to give two bands 13 and 17 kDa and cleaved parp gives one band at 90 kda. This is not what is shown in the figure.

>>Reply

The labelling of Fig.3E is correct. The manufacturer's website and multiple articles (doi: 10.1016/j.biopha.2022.113829; doi: 10.1186/s12944-021-01576-9; doi: 10.1007/s11418-023-01745-3) using the same antibody (Affinity, AF7022) show that cleaved caspase 3 only has one band at 17 KDa. we re-provided the WB images of cleaved PARP [Fig.3e].

Email: us@affinitytech.com 简体中文 | English | 日本語

Home Products & Services Promotions & News Research & Support Contact

Product: Cleaved-Caspase 3 (Asp175), p17 Antibody
Catalog: AF7022
Description: Rabbit polyclonal antibody to Cleaved-Caspase 3 (Asp175), p17
Application: WB IHC IF/ICC
Reactivity: Human, Mouse, Rat, Bovine
Prediction: Pig, Zebrafish, Bovine, Horse, Sheep, Rabbit, Dog, Xenopus
Mol.Wt.: 17kDa: 32kDa(Calculated)
Uniprot: P42574
RRID: AB_2855326

Size	Price	Inventory
50ul	\$250	In stock
100ul	\$350	In stock
200ul	\$450	In stock

Lead Time: Same day delivery
For pricing and ordering contact:
Local distributors

Related Downloads
MSDS
Data Sheet
COA

Protocols
WB Handbook
IHC Protocol
IF/ICC Protocol

8. Although this paper attempts to demonstrate that SH3RF2 is a critical factor in regulating cisplatin resistance. It is important that the authors demonstrate that in WT cells that SH3RF2 knockdown increases sensitivity of these cells to low levels of cisplatin. This may be a major argument to overcome chemotherapy toxicity if SH3RF3 therapy is ever implemented.

>>Reply

Thanks for your comments. The IC₅₀ value of DDP in the SH3RF2-silenced DDP-resistant cells was 3.7-~4.8-fold lower than that in WT DDP-resistant cells, indicating that SH3RF2 knockdown increased the sensitivity of these cells to DDP. Based on the present results, we speculated that SH3RF2 knockdown might also increase sensitivity

of parental cells to low levels of DDP. SH3RF2 was identified as an up-regulated gene in DDP-resistant cells when compared with the parental cells. Therefore, our study mainly focused on the effect of SH3RF2 on DDP resistance. We are very interested in the problem you point out. Our team will verify it in our future work. The limitation has been mentioned in the discussion section [the last paragraph].

9. In figure 5E sh3rf2 knockdown must be demonstrated.

>>Reply

The knockdown of SH3RF2 was verified by performing western blot analysis [Fig.6f].

10. In figure 5G RBPMS must be shown in the whole cell lysate. It is recommended the authors do this same experiment with overexpression of SH3RF2 and a catalytically inactive mutant of SH3RF2 to determine that the catalytic effect of SH3RF2 is critical for these effects and that SH3RF2 is not acting as a scaffold protein.

>>Reply

The expression of RBPMS in the whole cell lysate was shown in Figure 6i. We did the experiment you mentioned. As shown in Fig.6j, overexpression of the inactive mutant of SH3RF2 had no obvious effect on the ubiquitination of RBPMS. The results indicated that the catalytic effect of SH3RF2 was critical for the ubiquitination of RBPMS.

11. Figure 5f. Must be better explained. The reviewer does not understand anything in this figure. What does X mean? What does the circle mean? If X means knockdown shouldn't there be an increase in RBPMS. Should the same effect be seen in blot 3 as in block 1 as this is the same experiment.

>>Reply

The "circle" and "x" indicated whether the cells with stable expression shNC/sh-SH3RF2 were treated with CHX/MG132 or not. We modified Fig.6h to be well understood. The "x" was modified to "-". The circle was modified to "+".

Reviewer #3 (Remarks to the Author):

In 'SH3RF2 contributes to cisplatin resistance in ovarian cancer cells by promoting RBPMS degradation', the author innovatively identified the role of SH3RF2, an E3

ligase, in ovarian cancer (OC) cisplatin (DDP) resistance and explored the mechanisms it involved.

The authors have successfully established DDP-resistant cell lines based on A2780 and SKOV3 and reveal a significant increase of SH3RF2 in DDP-resistant cell line. Then knocked down SH3RF2 in the DDP-resistant cell lines was performed and demonstrated the lose of DDP-resistance in SH3RF2 knockdown cell lines Through PPI network analysis, RBPMS was identified. The effect of SH3RF2 in mediating K48-linked ubiquitination of RBPMS was demonstrated.

Major comments:

1. Cisplatin resistance is often related to cancer stemness. Have any change in stem-like properties or markers been observed.

>>Reply

Sphere formation assays were performed to analyze the stem-like properties of DDP-resistant cells. As shown in Fig.S2, the stem-like properties of DDP-resistant A2780 cells were significantly reduced by SH3RF2 silencing. SH3RF2 silencing had the tendency to decrease the stem-like properties of DDP-resistant SKOV3 cells.

2. Do RBPM2 and SH3RF2 colocalise? It seems RBPMS is cytoplasmic whereas SH3RF2 is nuclear (Fig. 5c). Where are they colocalized?

>>Reply

RBPMS-positive signals were located in the nucleus and the cytoplasm. SH3RF2-positive signals were located in the nucleus. The colocalization of SH3RF2 and RBPMS was shown in the nucleus.

3. A2780 has a paired drug resistant line. Has similar observation been seen using this paired cell line. It will be a more stringent test.

>>Reply

Thanks for your valuable comments. It is more rigorous that there are similar effects of SH3RF2 on cisplatin resistance in the paired cisplatin resistant cell line and the cisplatin resistant cell line established by ourselves. We'll purchase the paired cisplatin resistant cell line of A2780 and explore the effect of SH3RF2 on cisplatin resistance in our subsequent work.

4. The SH3RF2-RBPMS-AP-1 axis is not thoroughly tested in mice.

>>Reply

According to your comments, we detected SH3RF2 expression using IHC staining [Fig.5c], RBPMS expression using western blotting [Fig.6g], and *XIAP* and *MYC* expression levels in xenograft tumors using RT-qPCR analysis [Fig.7c&d].

5. Is there an association of SH3RF2-RBPMS-AP-1 axis with patients' samples?

>>Reply

We analyzed the correlation between SH3RF2 expression levels and clinicopathological parameters of ovarian cancer patients. The results showed that high expression of SH3RF2 was correlated with tumor sizes and cisplatin resistance in ovarian cancer patients [Table 1]. The expression of SH3RF2 was higher in cisplatin resistant patients than in cisplatin sensitive patients [Fig.1]. Conversely, the expression

of RBPMS was lower in cisplatin resistant patients than in cisplatin sensitive patients [Fig.S3]. Moreover, there was a negative correlation between SH3RF2 and RBPMS expression in ovarian cancer patients [Fig.6c].

REVIEWERS' COMMENTS:

Reviewer #1 (Remarks to the Author):

The authors answered my questions and the manuscript has improved. I have no further question.

Reviewer #2 (Remarks to the Author):

The authors have done a commendable job in answering the reviews questions. I have one minor issue concerning figure 6h. It is difficult to compare levels of RBPMs when the experiments are performed on different gels even though they are quantified. It is recommended that the experiment be performed on one gel. for ease 0 hrs vs 9 hrs.

the rest of the manuscript is suitable for publication.

Reviewer #3 (Remarks to the Author):

1. It is still difficult to confirm the nuclear localisation of RBPMs in the Figure and thus SH3RF2 and RBPMs colocalisation. Could subcellular fractionation and western blotting be used to confirm?
2. Please state the number of patient samples for Fig. 6c.

REVIEWERS' COMMENTS:

Reviewer #1 (Remarks to the Author):

The authors answered my questions and the manuscript has improved. I have no further question.

>>Reply

Thanks for your comments.

Reviewer #2 (Remarks to the Author):

The authors have done a commendable job in answering the reviews questions. I have one minor issue concerning figure 6h. It is difficult to compare levels of RBPMS when the experiments are performed on different gels even though they are quantified. It is recommended that the experiment be performed on one gel. for ease 0 hrs vs 9 hrs.

>>Reply

The experiment has been performed on one gel as suggested. The results were shown in Supplementary Fig.4c.

The rest of the manuscript is suitable for publication.

>>Reply

Thanks for your comments.

Reviewer #3 (Remarks to the Author):

1. It is still difficult to confirm the nuclear localisation of RBPMS in the Figure and thus SH3RF2 and RBPMS colocalisation. Could subcellular fractionation and western blotting be used to confirm?

>>Reply

According to your comments, the colocalization of RBPMS and SH3RF2 in the nucleus was verified by western blotting. The results in Supplementary Fig.4a revealed the expression of RBPMS and SH3RF2 in the nucleus.

2. Please state the number of patient samples for Fig. 6c.

>>Reply

The number of patient samples (n=120) was provided in the figure legends.